# Quantifying Statistical Significance of Neural Network-based Image Segmentation by Selective Inference

**Vo Nguyen Le Duy**
Nagoya Institute of Technology and RIKEN
duy.mllab.nit@gmail.com

**Shogo Iwazaki**
Nagoya Institute of Technology
iwazaki.s.mllab.nit@gmail.com

**Ichiro Takeuchi**[*]
Nagoya University and RIKEN
ichiro.takeuchi@mae.nagoya-u.ac.jp

## Abstract

Although a vast body of literature relates to image segmentation methods that use deep neural networks (DNNs), less attention has been paid to assessing the statistical reliability of segmentation results. In this study, we interpret the segmentation results as hypotheses driven by DNN (called DNN-driven hypotheses) and propose a method to quantify the reliability of these hypotheses within a statistical hypothesis testing framework. To this end, we introduce a conditional selective inference (SI) framework—a new statistical inference framework for data-driven hypotheses that has recently received considerable attention—to compute exact (non-asymptotic) valid $p$-values for the segmentation results. To use the conditional SI framework for DNN-based segmentation, we develop a new SI algorithm based on the homotopy method, which enables us to derive the exact (non-asymptotic) sampling distribution of DNN-driven hypothesis. We conduct several experiments to demonstrate the performance of the proposed method.

## 1 Introduction

Image segmentation is a fundamental task in computer vision, and it has been widely applied in many areas. The goal of image segmentation is to assign a label to every pixel in an observed image, such that pixels with the same label share certain characteristics. In the literature, numerous image segmentation algorithms have been developed based on thresholding [22], region-growing [21], or graph cuts [2, 3]. However, over the past few years, image segmentation using deep neural networks (DNNs) has become a popular model that exhibits remarkable performance improvements [19, 1, 25].

Although a vast body of literature relates to DNN-based segmentation methods, less attention has been paid to evaluating the statistical reliability of the segmentation results. In the absence of statistical reliability, it is difficult to manage the risk of obtaining incorrect segmentation results, which might be harmful when they are used in high-stakes decision-making, such as medical diagnoses or automatic driving. Therefore, it is necessary to develop a *valid* statistical inference method for data-driven DNN-based segmentation results that can properly control the risk of obtaining false positives.

Valid statistical inference on the DNN-based segmentation results is intrinsically difficult because the observed data is used twice: once for segmentation and once again for inference. This is often referred to as *double dipping* [14]. In statistics, it has been recognized that naively computing $p$-values in

---
[*]Corresponding author

double dipping is highly biased and correcting this bias is challenging, especially when the hypotheses are selected using complicated procedures such as DNN. Our idea is to introduce the *conditional Selective Inference (SI)* framework for resolving this challenge. The basic idea behind conditional SI is to perform statistical inference conditional on the selection event (i.e., segmentation results), which allows us to derive the exact (non-asymptotic) sampling distribution of the test statistic for making the valid statistical inference.

**Related works.** In the past few years, conditional SI has been recognized as a new promising approach by which to evaluate the statistical reliability of data-driven hypotheses, and it has been actively studied for inference on the features of linear models selected by several feature selection methods—for example, Lasso [17, 18, 15]. The basic idea of SI is to make inferences conditional on the selection event, which allows us to derive the exact sampling distribution of the test statistic. In addition, SI has been applied to various problems such as change point detection [32, 12, 13, 8, 33, 27], outlier detection [5, 31], generalized Lasso [11, 16], sequential feature selection [30, 26], and others [28, 23, 7, 10, 20, 6, 24]. However, no study to date provides conditional SI for DNNs.

The most closely related work (and the motivation for this study) is [29], where the authors provide a framework to compute valid $p$-values for image segmentation results obtained by threshold (TH)-based and graph cut (GC)-based segmentation algorithms. The novel idea of [29] is to consider the sampling distribution of the test statistic conditional on the selection event, in which the segmentation result is obtained. In that study, since the authors consider simple segmentation algorithms, the selection event is simply characterized by a set of linear or quadratic inequalities which is represented as a *single polytope* in the data space. Then, they can directly use the approach in the seminal conditional SI paper [17] to conduct the inference. However, it is not the case of DNN because the selection event of the DNN is much more complicated than that of TH and GC. Therefore, the method in [29] is *not* applicable in the case of DNN-based segmentation.

There are several types of problem setting with various network structures for DNN-based segmentation. In this study, we focus on the most standard one, the supervised segmentation setting with a simple structure (see §3 and §4). As will be described in §2, our target is to conduct the statistical inference in the test phase (not in the training phase). Namely, our goal is to quantify the reliability of the segmentation result when a new test image is given to a DNN that has been trained in advance.

**Contributions.** Our main contributions are as follows:

• To the best of our knowledge, this is the first study to provide an exact (non-asymptotic) inference method for statistically quantifying the reliability of image segmentation results obtained from DNNs.

• We propose a *homotopy method* to conduct a powerful and efficient conditional SI for DNN-based segmentation tasks. In this study, we mainly focus on a standard convolutional neural network (CNN) as a working example and show that all basic operations in the standard CNN (e.g., convolution, max pooling) can be incorporated in our proposed homotopy-based conditional SI method.

• We undertake experiments on both synthetic and real-world datasets, through which we offer evidence that our proposed method can successfully control the False Positive Rate (FPR), has good performance in terms of computational efficiency, and provides good results in practical applications.

Our implementation is available at

> https://github.com/vonguyenleduy/dnn_segmentation_selective_inference

## 2 Preliminary

In this section, we first review the conditional SI method for TH-based and GC-based segmentation algorithms in [29]. Then, we clarify the challenges of introducing the conditional SI for DNN-based segmentation tasks and present our idea to resolve the difficulties.

### 2.1 Conditional SI for TH-based and GC-based Image Segmentation [29]

Consider an image with $n$ pixels corrupted by Gaussian noise as

$$\boldsymbol{X} = (X_1, ..., X_n)^\top = \boldsymbol{\mu} + \boldsymbol{\varepsilon}, \quad \boldsymbol{\varepsilon} \sim \mathbb{N}(\boldsymbol{0}, \Sigma), \tag{1}$$

where $\boldsymbol{\mu} \in \mathbb{R}^n$ is an unknown mean pixel intensity vector, and $\boldsymbol{\varepsilon} \in \mathbb{R}^n$ is a vector of normally distributed noise with covariance matrix $\Sigma$, which is known or can be estimated from external data. We note that the assumption in Equation (1) does not mean that the pixel intensities in an image follow a multivariate normal distribution. Instead, the vector of noise added to the true pixel values are assumed to follows a multivariate normal distribution. For simplicity, we consider segmentation problems in which the image is divided into two regions: *object* and *background* regions [2]. We denote the sets of pixels in the object and background regions as $\mathcal{O}_{\boldsymbol{X}}$ and $\mathcal{B}_{\boldsymbol{X}}$, respectively.

**Statistical inference.** To quantify the statistical significance of the segmentation result, Tanizaki et al. [29] consider the following statistical test:

$$H_0 : \mu_{\mathcal{O}_{\boldsymbol{X}}} = \mu_{\mathcal{B}_{\boldsymbol{X}}} \quad \text{vs.} \quad H_1 : \mu_{\mathcal{O}_{\boldsymbol{X}}} \neq \mu_{\mathcal{B}_{\boldsymbol{X}}}, \tag{2}$$

where $\mu_{\mathcal{O}_{\boldsymbol{X}}}$ and $\mu_{\mathcal{B}_{\boldsymbol{X}}}$ are the true means of the pixel values in the object and background regions. A reasonable choice of the test statistic is the difference in the average pixel values between the object and background regions

$$\boldsymbol{\eta}^\top \boldsymbol{X} = \frac{1}{|\mathcal{O}_{\boldsymbol{X}}|} \sum_{i \in \mathcal{O}_{\boldsymbol{X}}} X_i - \frac{1}{|\mathcal{B}_{\boldsymbol{X}}|} \sum_{i \in \mathcal{B}_{\boldsymbol{X}}} X_i, \tag{3}$$

where $\boldsymbol{\eta} = \frac{1}{|\mathcal{O}_{\boldsymbol{X}}|} \mathbf{1}^n_{\mathcal{O}_{\boldsymbol{X}}} - \frac{1}{|\mathcal{B}_{\boldsymbol{X}}|} \mathbf{1}^n_{\mathcal{B}_{\boldsymbol{X}}}$ is the vector indicating the test-statistic direction, and $\mathbf{1}^n_{\mathcal{C}} \in \mathbb{R}^n$ is a vector whose elements belonging to a set $\mathcal{C}$ are set to 1, and 0 otherwise. We would like to note that the hypotheses in (2) and the test-statistic direction $\boldsymbol{\eta}$ in (3) are data-driven because they depend on $\boldsymbol{X}$. Given a significance level $\alpha \in [0, 1]$ (e.g., 0.05), we reject the null hypothesis $H_0$ if the $p$-value is smaller than $\alpha$, which indicates that the object region differs from the background region. Otherwise, we cannot say that a difference is significant.

**Conditional SI for computing valid $p$-values.** Let us define $\mathcal{A}(\boldsymbol{X})$ as the event in which the result of dividing the pixels of image $\boldsymbol{X}$ into the object region $\mathcal{O}_{\boldsymbol{X}}$ and the background region $\mathcal{B}_{\boldsymbol{X}}$ is obtained by applying a segmentation algorithm $\mathcal{A}$ (i.e., TH-based or GC-based image segmentation algorithm) on $\boldsymbol{X}$—that is,

$$\mathcal{A}(\boldsymbol{X}) = \{\mathcal{O}_{\boldsymbol{X}}, \mathcal{B}_{\boldsymbol{X}}\}. \tag{4}$$

To conduct a valid inference based on the concept of conditional SI, [29] proposes to consider the sampling distribution of the test statistic $\boldsymbol{\eta}^\top \boldsymbol{X}$ conditional on the segmentation results, that is,

$$\boldsymbol{\eta}^\top \boldsymbol{X} \mid \mathcal{A}(\boldsymbol{X}) = \mathcal{A}(\boldsymbol{x}^{\text{obs}}), \tag{5}$$

where $\boldsymbol{x}^{\text{obs}}$ is an observation (realization) of the random image $\boldsymbol{X}$. Then, to test the statistical significance of the segmentation result, the authors introduced the *selective $p$-value*, which satisfies the following sampling property:

$$\mathbb{P}_{H_0}(p_{\text{selective}} \leq \alpha \mid \mathcal{A}(\boldsymbol{X}) = \mathcal{A}(\boldsymbol{x}^{\text{obs}})) = \alpha, \tag{6}$$

i.e., $p_{\text{selective}}$ follows a uniform distribution under the null hypothesis. The $p_{\text{selective}}$ is defined as

$$p_{\text{selective}} = \mathbb{P}_{H_0}\left(|\boldsymbol{\eta}^\top \boldsymbol{X}| \geq |\boldsymbol{\eta}^\top \boldsymbol{x}^{\text{obs}}| \mid \mathcal{A}(\boldsymbol{X}) = \mathcal{A}(\boldsymbol{x}^{\text{obs}}), \boldsymbol{q}(\boldsymbol{X}) = \boldsymbol{q}(\boldsymbol{x}^{\text{obs}})\right), \tag{7}$$

where $\boldsymbol{q}(\boldsymbol{X})$ is the sufficient statistic of the nuisance parameter that needs to be conditioned on in order to tractably conduct the inference and it is defined as $\boldsymbol{q}(\boldsymbol{X}) = (I_n - \boldsymbol{c}\boldsymbol{\eta}^\top)\boldsymbol{X}$ with $\boldsymbol{c} = \Sigma\boldsymbol{\eta}(\boldsymbol{\eta}^\top \Sigma \boldsymbol{\eta})^{-1}$. Here, we note that the selective $p$-value depends on $\boldsymbol{q}(\boldsymbol{X})$, but the sampling property in (6) continues to be satisfied without this additional condition because we can marginalize over all values of $\boldsymbol{q}(\boldsymbol{X})$. The $\boldsymbol{q}(\boldsymbol{X})$ corresponds to the component $\boldsymbol{z}$ in the seminal conditional SI paper [17] (see Sec. 5, Eq. 5.2 and Theorem 5.2). We note that additional conditioning on $\boldsymbol{q}(\boldsymbol{X})$ is a standard approach in the conditional SI literature and is used in almost all the conditional SI-related studies.

**Example 1.** *(Selection event of TH-based segmentation and the conditional inference) Given a threshold $\tau$ and an observed data $\boldsymbol{x}^{\text{obs}}$, the selection event of TH-based segmentation is written as*

$$\mathcal{A}(\boldsymbol{X}) = \mathcal{A}(\boldsymbol{x}^{\text{obs}}) \Leftrightarrow \{\mathcal{O}_{\boldsymbol{X}}, \mathcal{B}_{\boldsymbol{X}}\} = \{\mathcal{O}_{\boldsymbol{x}^{\text{obs}}}, \mathcal{B}_{\boldsymbol{x}^{\text{obs}}}\} \Leftrightarrow \left\{ \boldsymbol{X} \in \mathbb{R}^n : \begin{array}{l} X_i \geq \tau, \; \forall i \in \mathcal{O}_{\boldsymbol{x}^{\text{obs}}}, \\ X_j < \tau, \; \forall j \in \mathcal{B}_{\boldsymbol{x}^{\text{obs}}} \end{array} \right\},$$

*which is simply a single polytope in the data space. Then, the conditional inference in (5) for the case of TH-based segmentation is explicitly defined as*

$$\boldsymbol{\eta}^\top \boldsymbol{X} \mid \{X_i \geq \tau, \; \forall i \in \mathcal{O}_{\boldsymbol{x}^{\text{obs}}}, X_j < \tau, \; \forall j \in \mathcal{B}_{\boldsymbol{x}^{\text{obs}}}\}.$$

---

[2]The extension to the cases where the image is divided into more than two regions is straightforward.

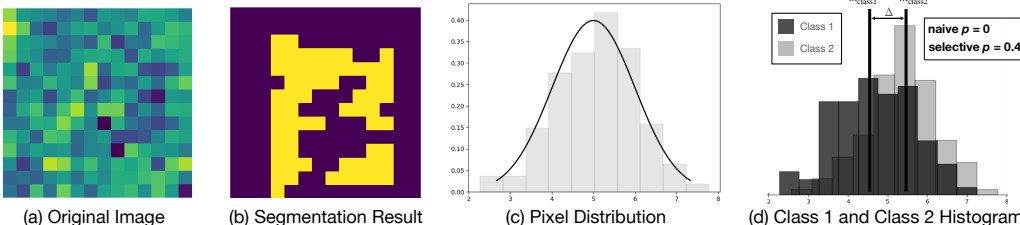

|(a) Original Image|(b) Segmentation Result|(c) Pixel Distribution|(d) Class 1 and Class 2 Histogram|

Figure 1: Illustration of the bias in naive statistical inference. (a) We generate a null image in which all the "true" pixel values are the same. (b) The segmentation result obtained by a trained CNN. (c) The distribution and the histogram of the pixel intensities. (d) The histograms of the pixel intensities in the background (Class 1) and object (Class 2), as well as the naive $p$-value and the proposed selective $p$-value. Even from an image containing no object, the naive $p$-value is very small, indicating that it cannot be used to evaluate the reliability of the segmentation result. By applying the proposed method, we can successfully identify the false segmentation result.

Similarly, [29] showed that the selection event of GC-based segmentation can be approximated by a set of quadratic inequalities which is also represented by a polytope. When the selection event is represented by a single polytope, we can directly apply the original conditional SI method by [17].

## 2.2 Challenges of Introducing Conditional SI for DNN-based Image Segmentation

In this paper, we consider the same problem setup as described in [29] (i.e., from (2) to (7)). The main difference is that the segmentation event $\mathcal{A}(\boldsymbol{X}) = \{\mathcal{O}_{\boldsymbol{X}}, \mathcal{B}_{\boldsymbol{X}}\}$ in (4) is now obtained from a DNN that has already been trained in advance for the segmentation task. The discussion thus far indicates that we can conduct an exact statistical inference for the segmentation result if we can compute the selective $p$-value in (7). However, to compute $p_{\text{selective}}$, the major challenge is to characterize selection event $\{\mathcal{A}(\boldsymbol{X}) = \mathcal{A}(\boldsymbol{x}^{\text{obs}})\}$ of DNN-based segmentation, which is much more complicated and cannot be simply represented by an intersection of linear or quadratic inequalities as in the case of TH-based and GC-based segmentations. In the next section, we propose a novel method that addresses all the aforementioned challenges by utilizing the concept of a parametrized line search (i.e., the homotopy method). Figure 1 illustrates the selection bias in naive statistical inference and the importance of the proposed method.

## 3 Proposed Method

In this section, we propose a method for computing the selective $p$-values in (7). The main task is to identify the following set of $\boldsymbol{x} \in \mathbb{R}^n$ that satisfies the condition part:

$$\mathcal{X} = \{\boldsymbol{x} \in \mathbb{R}^n \mid \mathcal{A}(\boldsymbol{x}) = \mathcal{A}(\boldsymbol{x}^{\text{obs}}), \boldsymbol{q}(\boldsymbol{x}) = \boldsymbol{q}(\boldsymbol{x}^{\text{obs}})\}. \tag{8}$$

To identify $\mathcal{X}$, we first reformulate the problem of identifying this data space as a problem of searching the data on a parametrized line. Then, we introduce a homotopy method to conduct a parametrized line search illustrated in Fig. 2.

### 3.1 Characterization of the Conditional Data Space $\mathcal{X}$

In (8), according to the second condition on the sufficient statistic of the nuisance parameter $\boldsymbol{q}(\boldsymbol{x})$, the data in $\mathcal{X}$ is restricted to a line in $\mathbb{R}^n$, as stated in the following lemma.

**Lemma 1.** *Let us define*

$$\boldsymbol{a} = \boldsymbol{q}(\boldsymbol{x}^{\text{obs}}) \quad and \quad \boldsymbol{b} = \Sigma\boldsymbol{\eta}(\boldsymbol{\eta}^\top\Sigma\boldsymbol{\eta})^{-1}. \tag{9}$$

*Then, the conditional data space $\mathcal{X}$ can be rewritten using a scalar parameter $z \in \mathbb{R}$ as*

$$\mathcal{X} = \{\boldsymbol{x}(z) = \boldsymbol{a} + \boldsymbol{b}z \mid z \in \mathcal{Z}\}, \quad where \quad \mathcal{Z} = \{z \in \mathbb{R} \mid \mathcal{A}(\boldsymbol{x}(z)) = \mathcal{A}(\boldsymbol{x}^{\text{obs}})\}. \tag{10}$$

*Proof.* The proof is deferred to Appendix A. ∎

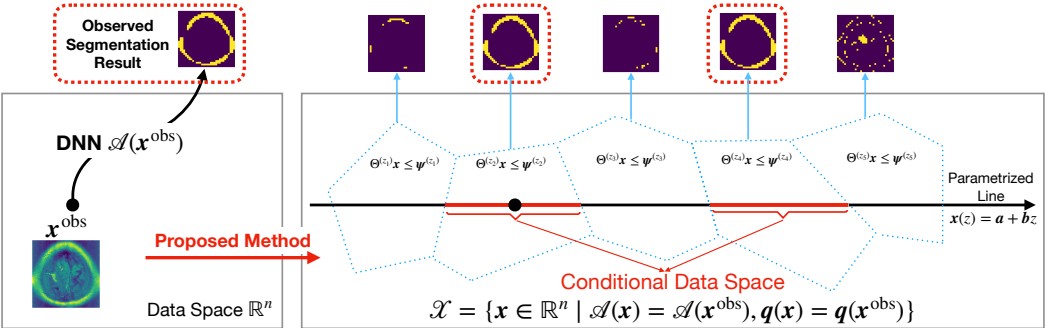

Figure 2: A schematic illustration of the proposed method. By applying a trained CNN to the observed image $\boldsymbol{x}^{\mathrm{obs}}$, we obtain a segmentation result. Then, we parametrize $\boldsymbol{x}^{\mathrm{obs}}$ with a scalar parameter $z$ in the dimension of the test statistic to identify the subspace $\mathcal{X}$ whose data have the *same* segmentation result as $\boldsymbol{x}^{\mathrm{obs}}$. Finally, the valid statistical inference is conducted conditional on $\mathcal{X}$. We introduce a homotopy method for efficiently characterizing the conditional data space $\mathcal{X}$.

We note that whereas the seminal conditional SI work [17] already implicitly considers the data on the line, this was explicitly discussed for the first time in Section 6 of [18]. Lemma 1 indicates that we need not consider the $n$-dimensional data space. Instead, we need only consider the *one-dimensional projected* data space $\mathcal{Z}$ in (10). Now, let us consider a random variable $Z \in \mathbb{R}$ and its observation $z^{\mathrm{obs}} \in \mathbb{R}$ that satisfies $\boldsymbol{X} = \boldsymbol{a} + \boldsymbol{b}Z$ and $\boldsymbol{x}^{\mathrm{obs}} = \boldsymbol{a} + \boldsymbol{b}z^{\mathrm{obs}}$. The selective $p$-value (7) is rewritten as

$$
\begin{aligned}
p_{\mathrm{selective}} &= \mathbb{P}_{\mathrm{H}_0}\left(|\boldsymbol{\eta}^\top \boldsymbol{X}| \geq |\boldsymbol{\eta}^\top \boldsymbol{x}^{\mathrm{obs}}| \mid \boldsymbol{X} \in \mathcal{X}\right) \\
&= \mathbb{P}_{\mathrm{H}_0}\left(|Z| \geq |z^{\mathrm{obs}}| \mid Z \in \mathcal{Z}\right).
\end{aligned}
\tag{11}
$$

Because the variable $Z \sim \mathbb{N}(0, \boldsymbol{\eta}^\top \Sigma \boldsymbol{\eta})$ under the null hypothesis, $Z \mid Z \in \mathcal{Z}$ follows a *truncated* normal distribution. Once the truncation region $\mathcal{Z}$ is identified, computation of the selective $p$-value in (11) is straightforward. Therefore, the remaining task is to identify $\mathcal{Z}$.

## 3.2 Identification of Truncation Region $\mathcal{Z}$

As discussed in §3.1, to calculate the selective $p$-value (11), we must identify the truncation region $\mathcal{Z}$ in (10). To construct $\mathcal{Z}$, we must (a) compute $\mathcal{A}(\boldsymbol{x}(z))$ for all $z \in \mathbb{R}$, and (b) identify the set of intervals of $z$ on which $\mathcal{A}(\boldsymbol{x}(z)) = \mathcal{A}(\boldsymbol{x}^{\mathrm{obs}})$. However, it seems intractable to obtain $\mathcal{A}(\boldsymbol{x}(z))$ for infinitely many values of $z \in \mathbb{R}$.

Our first idea of introducing conditional SI for DNN is that we additionally condition on some extra events to make the problem tractable. We now focus on a class of DNNs whose activation functions (AFs) are piecewise-linear—for example, ReLU, Leaky ReLU. Then, we consider additional conditioning on the selected piece of each piecewise-linear AF.

**Definition 1.** *Let $s_j(\boldsymbol{x})$ be "the selected piece" of a piecewise-linear AF at the $j$-th unit in a DNN for a given input image $\boldsymbol{x}$, and let $\boldsymbol{s}(\boldsymbol{x})$ be the set of $s_j(\boldsymbol{x})$ for all the nodes in a DNN.*

For example, given a ReLU AF, $s_j(\boldsymbol{x})$ takes either 0 or 1, depending on whether the input to the $j$-th unit is located at the flat part (inactive) or the linear part (active) of the ReLU function. Using the notion of selected pieces $\boldsymbol{s}(x)$, instead of computing the selective $p$-value in (11), we consider the following *over-conditioning (oc)* conditional $p$-value:

$$
p_{\mathrm{selective}}^{\mathrm{oc}} = \mathbb{P}_{\mathrm{H}_0}\left(|Z| \geq |z^{\mathrm{obs}}| \mid Z \in \mathcal{Z}^{\mathrm{oc}}\right),
\tag{12}
$$

where

$$
\mathcal{Z}^{\mathrm{oc}} = \left\{z \in \mathbb{R} \mid \mathcal{A}(\boldsymbol{x}(z)) = \mathcal{A}(\boldsymbol{x}^{\mathrm{obs}}), \boldsymbol{s}(\boldsymbol{x}(z)) = \boldsymbol{s}(\boldsymbol{x}^{\mathrm{obs}})\right\}.
$$

However, such an over-conditioning in SI leads to a loss of statistical power [17, 9].

Our second idea is to develop a *homotopy method* to resolve the over-conditioning problem—that is, remove the conditioning of $\boldsymbol{s}(\boldsymbol{x}(z)) = \boldsymbol{s}(\boldsymbol{x}^{\mathrm{obs}})$. Using the homotopy method, we can efficiently compute $\mathcal{A}(\boldsymbol{x}(z))$ in a finite number of operations without the need to consider infinitely many values of $z \in \mathbb{R}$, which is subsequently used to obtain the truncation region $\mathcal{Z}$ in (10). The main idea is to

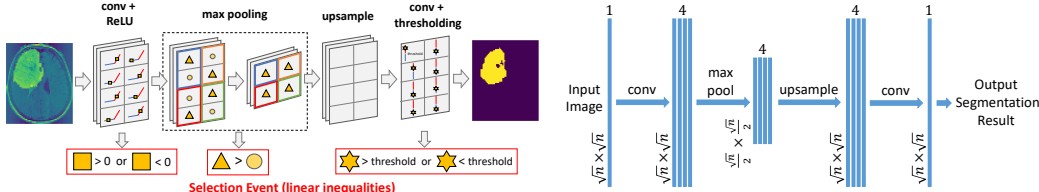

(a) The over-conditioning selection event of a basic CNN.  (b) Basic network structure for segmentation.

Figure 3: (a) The over-conditioning selection event of a basic CNN can be written as a set of linear inequalities because the basic operations (e.g., convolution, max pooling) and activation functions (e.g., ReLU, thresholding) of a CNN can be represented as linear inequalities. Note that non-piecewise linear functions such as sigmoid or hyperbolic tangent function are equivalent to consider the values prior to activation with threshold value 0. (b) Basic network structure for segmentation.

compute a finite number of *breakpoints* at which one node of the network changes its status from active to inactive or vice versa. This concept is similar to the regularization path of Lasso, where we can compute a finite number of breakpoints at which the active set changes.

To this end, we introduce a two-step iterative approach, generally described as follows (see Fig. 2):

• **Step 1 (over-conditioning step).** Considering the over-conditioning case by additionally conditioning on the selected pieces of all hidden nodes in the DNN.

• **Step 2 (homotopy step).** Combining multiple over-conditioning cases using the homotopy method to obtain $\mathcal{A}(\boldsymbol{x}(z))$ for all $z \in \mathbb{R}$.

### 3.3 Step 1: Over-Conditioning Step

We show that by additionally conditioning on the selected pieces $\boldsymbol{s}(\boldsymbol{x}^{\mathrm{obs}})$ of all the hidden nodes, the selection event of DNN is written as a set of linear inequalities. The illustration is shown in Fig. 3a.

**Lemma 2.** *Consider a class of DNN that consists of affine operations and piecewise-linear AFs. Then, the overconditioning region is written as*

$$\mathcal{Z}^{\mathrm{oc}} = \{z \in \mathbb{R} \mid \Theta^{(\boldsymbol{s}(\boldsymbol{x}^{\mathrm{obs}}))}\boldsymbol{x}(z) \leq \boldsymbol{\psi}^{(\boldsymbol{s}(\boldsymbol{x}^{\mathrm{obs}}))}\}$$

*using a matrix $\Theta^{(\boldsymbol{s}(\boldsymbol{x}^{\mathrm{obs}}))}$ and a vector $\boldsymbol{\psi}^{(\boldsymbol{s}(\boldsymbol{x}^{\mathrm{obs}}))}$ that depend only on the selected pieces $\boldsymbol{s}(\boldsymbol{x}^{\mathrm{obs}})$.*

*Proof.* For the class of DNN that consists of affine operations and piecewise-linear AFs, by fixing the selected pieces of all the piecewise-linear AFs, the input to each AF is represented by an affine function of an image $\boldsymbol{x}$. Therefore, the condition for selecting a piece in a piecewise-linear AF, $s_j(\boldsymbol{x}(z)) = s_j(\boldsymbol{x}^{\mathrm{obs}})$, is written as a linear inequality with respect to $\boldsymbol{x}(z)$. ∎

**Remark 1.** (Activation functions in the hidden layers) In this work, we focus on a trained DNN where the AFs used at hidden layers are *piecewise linear*—for example, ReLU, Leaky ReLU, which is commonly used in a CNN. Otherwise, if there is any specific demand to use non-piecewise linear functions (such as sigmoid or hyperbolic tangent), we can apply a piecewise-linear approximation approach to these functions. We provide examples of such approximation in Appendix D.

**Remark 2.** (Operations in a trained neural network) Most basic operations in a trained neural network are written as affine operations. In a traditional neural network, the multiplication results between the weight matrix and the output of the previous layer and its summation with the bias vector is an affine operation. In a CNN, the convolution and upsampling operations are obviously affine operations. Even when an operation in a DNN is NOT represented as a *single* affine operation, it is often the case that the operation is represented by as a *set* of affine operations, as in the following remark.

**Remark 3.** (Max-pooling) Although the max-pooling is not an affine operation, it can be written as a set of linear inequalities. For instance, $v_1 = \max\{v_1, v_2, v_3\}$ can be written as a set $\{\boldsymbol{e}_1^\top \boldsymbol{v} \leq \boldsymbol{e}_2^\top \boldsymbol{v}, \boldsymbol{e}_1^\top \boldsymbol{v} \leq \boldsymbol{e}_3^\top \boldsymbol{v}\}$, where $\boldsymbol{v} = (v_1, v_2, v_3)^\top$ and $\boldsymbol{e}_i$ is the standard basis vector with 1 at position $i$.

**Remark 4.** (Activation functions at the output layer) In Remark 1, we mention that we need to perform piecewise linear approximation for non-piecewise linear activations. However, if these

| **Algorithm 1** `parametric_SI_DNN` | **Algorithm 2** `compute_solution_path` |
|---|---|
| **Input:** $\boldsymbol{x}^{\text{obs}}, z_{\min}, z_{\max}$ | **Input:** $\boldsymbol{a}, \boldsymbol{b}, z_{\min}, z_{\max}$ |
| 1: Obtain $\mathcal{A}(\boldsymbol{x}^{\text{obs}})$ by applying the trained DNN to $\boldsymbol{x}^{\text{obs}}$ | 1: Initialization: $t = 1, z_t = z_{\min}, \mathcal{T} = \{z_t\}$ |
| 2: Compute $\boldsymbol{\eta} \leftarrow$ Eq. (3) | 2: **while** $z_t < z_{\max}$ **do** |
| 3: Calculate $\boldsymbol{a}$ and $\boldsymbol{b} \leftarrow$ Eq. (9) | 3:    Compute $\boldsymbol{x}(z_t) = \boldsymbol{a} + \boldsymbol{b}z_t$ in $\mathbb{R}^n$ of $z_t$ |
| 4: $\mathcal{A}(\boldsymbol{x}(z)) \leftarrow$ `compute_solution_path`$(\boldsymbol{a}, \boldsymbol{b}, z_{\min}, z_{\max})$ // Algorithm 2 | 4:    Obtain $\mathcal{A}(\boldsymbol{x}(z_t))$ by applying a trained DNN to $\boldsymbol{x}(z_t)$ |
| 5: Identify $\mathcal{Z} \leftarrow \{z : \mathcal{A}(\boldsymbol{x}(z)) = \mathcal{A}(\boldsymbol{x}^{\text{obs}})\}$ | 5:    Compute $z_{t+1} \leftarrow$ Lemma 3 |
| 6: $p_{\text{selective}} \leftarrow$ Eq. (11) | 6:    $\mathcal{T} = \mathcal{T} \cup \{z_{t+1}\}$, and $t = t + 1$ |
| **Output:** $p_{\text{selective}}$ | 7: **end while** |
|  | **Output:** $\{\mathcal{A}(\boldsymbol{x}(z_t))\}_{z_t \in \mathcal{T}}$ |

functions are used at the output layer (e.g., sigmoid function or tanh function), we need *not* perform the approximation task, because we can define the set of linear inequalities based on the values prior to activation. See the next Example 2 for the case of a sigmoid function.

The list of components that are commonly used in a DNN for segmentation task and can be represented as a set of linear inequalities is provided in Appendix E. We would like to note that, since we are primarily interested in the reliability of a trained network given new inputs (i.e., not training inputs), the validity of our proposed method does not depend on DNN training. Therefore, all of the operations that only operate in the training process such as Dropout do not affect the applicability of the proposed SI approach and we do not need consider the selection event of those operations in our method.

**Example 2.** (The selection event of a basic CNN is a set of linear inequalities) Consider a basic network structure for segmentation as shown in Fig. 3b and $n$ is an even number. Let us start with the first convolutional layer with 4 filters. Let $R^f \in \mathbb{R}^{\sqrt{n} \times \sqrt{n}}$ be the matrix obtained by the $f^{\text{th}}$ filter. We apply ReLU activation on $R^f, \forall f \in [4]$. The selection event of ReLU is written as

$$
\mathcal{S}_{\text{ReLU}} = \bigcup_{f=1}^{4} \bigcup_{i=1}^{\sqrt{n}} \bigcup_{j=1}^{\sqrt{n}} \left\{ \begin{array}{l} R_{ij}^f \geq 0, \text{ if output of ReLU} \geq 0, \\ R_{ij}^f < 0, \text{ otherwise.} \end{array} \right\}
$$

Let $P^f \in \mathbb{R}^{\sqrt{n} \times \sqrt{n}}$ is the matrix after applying ReLU on $R^f$. We now move to the pooling layer by applying the max-pooling on $P^f, \forall f \in [4]$, with $2 \times 2$ window. Selection event of max-pooling is

$$
\mathcal{S}_{\text{MaxPool}} = \bigcup_{f=1}^{4} \bigcup_{i \in \mathcal{I}} \bigcup_{j \in \mathcal{I}} \left\{ P_{ij}^f \leq q, \ P_{i(j+1)}^f \leq q, P_{(i+1)j}^f \leq q, \ P_{(i+1)(j+1)}^f \leq q \right\},
$$

where $q = \max\{P_{ij}^f, P_{i(j+1)}^f, P_{(i+1)j}^f, P_{(i+1)(j+1)}^f\}$ and $\mathcal{I} = \{2\kappa + 1 : \kappa \in \{0, 1, ..., (n/2 - 1)\}\}$. We continue the forward process until the last convolutional layer. Let $W \in \mathbb{R}^{\sqrt{n} \times \sqrt{n}}$ be the matrix at this final layer. We apply sigmoid function on $W$. The selection event of sigmoid is written as

$$
\mathcal{S}_{\text{Sigmoid}} = \bigcup_{i=1}^{\sqrt{n}} \bigcup_{j=1}^{\sqrt{n}} \left\{ \begin{array}{l} W_{ij} \geq 0, \text{ if sigmoid output} \geq 0.5, \\ W_{ij} < 0, \text{ otherwise.} \end{array} \right\}
$$

Finally, the over-conditioning selection event is

$$
\mathcal{S} = \mathcal{S}_{\text{ReLU}} \bigcup \mathcal{S}_{\text{MaxPool}} \bigcup \mathcal{S}_{\text{Sigmoid}},
$$

which is a set of linear inequalities.

### 3.4   Step 2: Homotopy Step

We introduce homotopy method to compute $\mathcal{A}(\boldsymbol{x}(z))$ by combining multiple over-conditioning steps.

**Lemma 3.** *Consider a real value $z_t$. By applying a trained DNN to $\boldsymbol{x}(z_t)$, we obtain a set of linear inequalities $\Theta^{(\boldsymbol{s}(\boldsymbol{x}(z_t)))} \boldsymbol{x}(z_t) \leq \boldsymbol{\psi}^{(\boldsymbol{s}(\boldsymbol{x}(z_t)))}$. Then, the next breakpoint $z_{t+1} > z_t$, at which the status of one node is changed from active to inactive or vice versa—that is, the sign of one linear inequality changes. This breakpoint is calculated by*

$$
z_{t+1} = \min_{k:(\Theta^{(\boldsymbol{s}(\boldsymbol{x}(z_t)))} \boldsymbol{b})_k > 0} \frac{\psi_k^{(\boldsymbol{s}(\boldsymbol{x}(z_t)))} - (\Theta^{(\boldsymbol{s}(\boldsymbol{x}(z_t)))} \boldsymbol{a})_k}{(\Theta^{(\boldsymbol{s}(\boldsymbol{x}(z_t)))} \boldsymbol{b})_k}.
$$

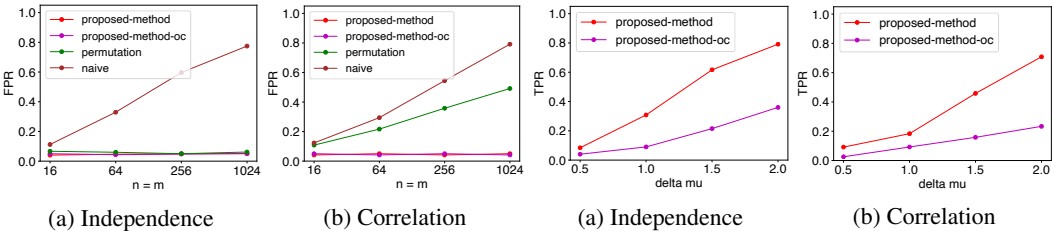

| (a) Independence | (b) Correlation | (a) Independence | (b) Correlation |

Figure 4: False Positive Rate (FPR) comparison.  Figure 5: True Positive Rate (TPR) comparison.

The proof of Lemma 3 is provided in Appendix B. Algorithm 2 shows our solution to efficiently identify $\mathcal{A}(\boldsymbol{x}(z))$. In this algorithm, multiple *breakpoints* $z_1 < z_2 < ... < z_{|\mathcal{T}|}$ are computed one by one. Each breakpoint $z_t, t \in [|\mathcal{T}|]$ indicates the point at which the sign of one linear inequality is changed, i.e., the status of one node in the network will change from active to inactive or vice versa. By identifying all these breakpoints $\{z_t\}_{t \in [|\mathcal{T}|]}$, the solution path is given by $\mathcal{A}(\boldsymbol{x}(z)) = \mathcal{A}(\boldsymbol{x}(z_t))$ if $z \in [z_t, z_{t+1}], t \in [|\mathcal{T}|]$. Algorithm 1 shows the entire procedure to calculate the selective $p$-value in (11). First, we apply a DNN on $\boldsymbol{x}^{\mathrm{obs}}$ to obtain the segmentation result $\mathcal{A}(\boldsymbol{x}^{\mathrm{obs}})$. Next, we calculate $\boldsymbol{\eta}$ which is used to identify a parameterized line in $\mathbb{R}^n$. Then, we compute $\mathcal{A}(\boldsymbol{x}(z))$ by using Algorithm 2. After obtaining $\mathcal{A}(\boldsymbol{x}(z))$, we identify $\mathcal{Z}$ in (10), which is the key factor for computing the selective $p$-value. Regarding the choice of $[z_{\min}, z_{\max}]$, under a normal distribution, neither very positive nor very negative $z$ values affect the inference. Therefore, it is reasonable to consider a range of values, e.g., $[-20\sigma, 20\sigma]$ [18], where $\sigma$ is the standard deviation of the distribution of the test statistic.

**Complexity.**    In the literature of homotopy method (a.k.a. parametric programming), it is known that the actual computational cost differs significantly from the worst case. A well-known application of the homotopy method in the ML community is the Lasso regularization path, which also has the worst-case computational cost on the exponential order of the number of features, but the actual cost is known to be nearly linear order. Similarly, in the proposed homotopy method for SI, it is also evident from our experimental results that the number of breakpoints is almost linearly increasing in practice (Fig. 6). We additionally note that, at step 4 of Algorithm 2, we apply a trained DNN to the parametrized data $\boldsymbol{x}(z_t)$ at any breakpoints $z_t$. Therefore, the number of forward passes of the network is equal to the number of breakpoints.

**Class of networks.**    The proposed method can be applied to a class of *piecewise linear networks* whose network operations are characterized by a set of linear inequalities or approximated by piecewise-linear functions (many state-of-the-art image segmentation networks are or can be well-approximated by piecewise linear networks). This is due to the reason that all the theories and algorithms in §3 only depend on the property of each component and does not depend on the entire structure of the network. We believe that our method is fairly general because most of the basic operations in a trained neural network can be decomposed or approximated by affine operations as we presented in Remarks 1-4, Example 2 as well as Appendix D.

## 4   Experiment

**Experimental setup.** We compared the proposed method (homotopy method) with proposed-method-oc (over-conditioning), naive method and permutation test. The details of the methods for comparison are shown in Appendix F. We considered two covariance matrices:

- $\Sigma = I_n$ (independence) and
- $\Sigma = [0.5^{|i-j|}]_{ij} \in \mathbb{R}^{n \times n}$ (correlation).

Regarding the FPR experiments, we generated 120 null images $\boldsymbol{X} = (X_1, ..., X_n) \sim \mathbb{N}(\boldsymbol{0}_n, \Sigma)$ for $n \in \{16, 64, 256, 1024\}$. For each value of $n$, we run 120 trials. To test the power, we generated images $\boldsymbol{X}$ with $n = 256$, in which the *true* average difference in the underlying model $\mu_{\mathcal{O}_{\boldsymbol{x}}} - \mu_{\mathcal{B}_{\boldsymbol{x}}} = \Delta_\mu \in \{0.5, 1.0, 1.5, 2.0\}$. For each $\Delta_\mu$, we run 120 trials. We selected the significance level $\alpha = 0.05$ and we used the basic CNN shown in Fig. 3b.

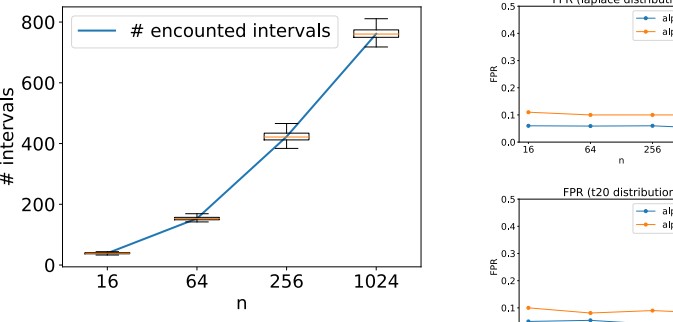

Figure 6: The number of encountered intervals on the line along the direction of the test statistic is almost linearly increasing in practice.

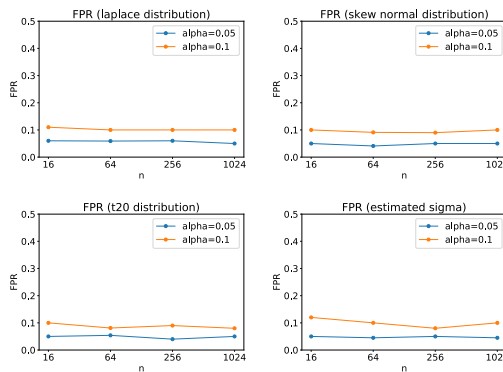

Figure 7: Robustness of the proposed method.

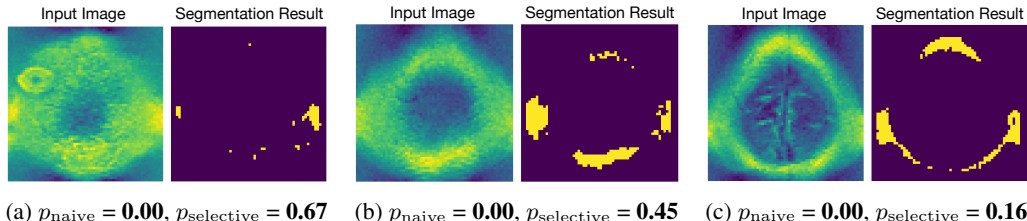

(a) $p_{\mathrm{naive}} = \mathbf{0.00}$, $p_{\mathrm{selective}} = \mathbf{0.67}$ (b) $p_{\mathrm{naive}} = \mathbf{0.00}$, $p_{\mathrm{selective}} = \mathbf{0.45}$ (c) $p_{\mathrm{naive}} = \mathbf{0.00}$, $p_{\mathrm{selective}} = \mathbf{0.16}$

Figure 8: Inference on segmentation results for images without a tumor region.

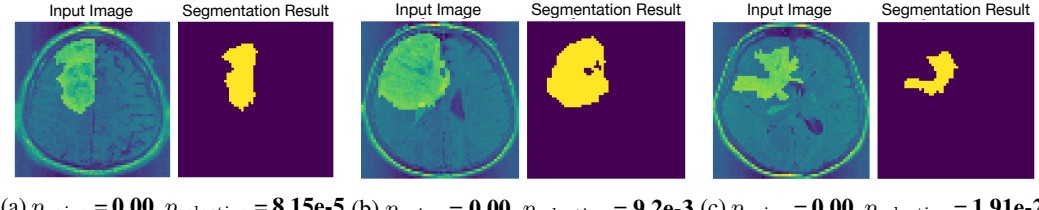

(a) $p_{\mathrm{naive}} = \mathbf{0.00}$, $p_{\mathrm{selective}} = \mathbf{8.15e\text{-}5}$ (b) $p_{\mathrm{naive}} = \mathbf{0.00}$, $p_{\mathrm{selective}} = \mathbf{9.2e\text{-}3}$ (c) $p_{\mathrm{naive}} = \mathbf{0.00}$, $p_{\mathrm{selective}} = \mathbf{1.91e\text{-}2}$

Figure 9: Inference on segmentation results for images where there exists a tumor region.

In the literature, many network architectures have been proposed for segmentation problems. Although these architectures are very complex, if they are broken down into smaller elements, each can be represented (or at least well approximated) as an affine operation whose selection event can be handled by the proposed method. Because the main contribution of this study is to introduce conditional SI by which to evaluate the reliability of the segmentation results, empirical implementations and evaluations of the complex architectures are beyond the scope of this study.

**Numerical results.** The results of the FPR control are shown in Fig. 4. The proposed methods successfully control the FPR in both cases of independence and correlation while the naive and permutation methods *cannot*. Because the naive and permutation methods fail to control the FPR, we no longer consider the power. Fig. 5 shows that the homotopy method has higher TPR than the over-conditioning option. Fig. 6 shows the reason why the proposed homotopy method is efficient. Intuitively, to overcome the over-conditioning issue, we need to consider all combinations of the activenesses of the nodes in a trained DNN, which is exponentially increasing. With the homotopy method, we need only consider the number of encountered intervals on the line along the direction of the test statistic, which is almost linearly increasing in practice. Additionally, we confirm the robustness of the proposed method in terms of FPR control by applying our method to data following Laplace distribution, skew normal distribution (skewness coefficient 10), and $t_{20}$ distribution. We also consider the case where variance is also estimated from the data. We test the FPR for both $\alpha = 0.05$ and $\alpha = 0.1$. The FPR results are shown in Fig. 7. Our method still maintains good performance. Regarding the last plot (estimated sigma) in Fig. 7, since we want to empirical check the performance

Table 1: False positive rate and power comparisons in brain image dataset.

|  | **FPR** | **Power** |
|---|---|---|
| **Proposed Method** | 0.057 | 0.683 |
| **Permutation Test** | 0.640 (unreliable) | N.A |

on FPR control when breaking the assumption of $\Sigma$ (i.e., $\Sigma$ is known or estimated from independent data), we set $\Sigma = s^2 I$ where $s^2$ is the empirical (sample) variance and $I$ is the identity matrix.

**Real-data examples.** We examine the brain image dataset extracted from the dataset used in [4], which includes 939 images with tumors and 941 images without tumors. We first compare our method to a permutation test in terms of FPR control. The results are shown in Table 1. Because the permutation test cannot properly control the FPR, a comparison of power is no longer needed. Comparisons of the naive $p$-value and the selective $p$-value are shown in Fig. 8 and Fig. 9. The naive $p$-value remains small, even when the image has no tumor region; this indicated that naive $p$-values cannot be used to quantify the reliability of DNN-based segmentation results. The proposed method successfully identifies false positive and true positive detections.

## 5 Discussion

We propose a novel conditional SI method to conduct inference on the DNN-based image segmentation result. We believe that this study stands as a significant step toward reliable artificial intelligence and opens several directions for statistically evaluating the reliability of DNN-driven hypotheses. Some open questions remain, as follows:

• The proposed method currently does not support softmax and batch normalization operators. Although these operators can in theory be approximated by affine operations, doing so will be challenging in reality, as we need to work in a multidimensional space. Although it is difficult to apply the proposed method to network architectures containing components that are difficult to decompose into affine operations, one possible solution is to devise a new network architecture that can be expressed as a set of affine functions but still has state-of-the-art segmentation performance.

• Although there is no technical limitation in applying our proposed method to state-of-the-art network structures, it requires a considerable amount of implementation effort. This is because, for a large and complex network, we have to implement the selection events for all the components in the network. The future direction could be first to implement the proposed method for a few commonly used network structures, and then to develop a generic software tool that can automatically compute the selection event when the network structure is given.

• Even though this study mainly focuses on the image segmentation task, the proposed method can be applied to a class of problems where (1) the test statistic is defined as a linear contrast w.r.t the data and (2) the operations of the trained neural network can be characterized by a set of linear inequalities (or approximated by piecewise-linear functions). Therefore, widening the applicability of the proposed method to other computer vision tasks—as well as to other fields such as natural language processing and signal processing—would also stand as a valuable future contribution.

## Acknowledgments and Disclosure of Funding

We thank the anonymous reviewers and area chair for their comments. This work was partially supported by MEXT KAKENHI (20H00601), JST CREST (JPMJCR21D3), JST Moonshot R&D (JPMJMS2033-05), JST AIP Acceleration Research (JPMJCR21U2), NEDO (JPNP18002, JPNP20006), RIKEN Center for Advanced Intelligence Project, and RIKEN Junior Research Associate Program.

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
