# A    Proof of Lemma 1

According to the second condition in (8), we have

$$q(x) = q(x^{\mathrm{obs}})$$
$$\Leftrightarrow \left(I_n - c\eta^\top\right)x = q(x^{\mathrm{obs}})$$
$$\Leftrightarrow x = q(x^{\mathrm{obs}}) + \frac{\Sigma\eta}{\eta^\top\Sigma\eta}\eta^\top x.$$

By defining $a = q(x^{\mathrm{obs}})$, $b = \Sigma\eta(\eta^\top\Sigma\eta)^{-1}$, $z = \eta^\top x$, $x(z) = a + bz$, and incorporating the first condition in (8), we obtain the result in Lemma 1.

# B    Proof of Lemma 3

Given a real value $z_t$, we can always obtain $\Theta^{(s(x(z_t)))}x(z_t) \leq \psi^{(s(x(z_t)))}$ when applying the trained neural network on $x(z_t)$. Then, for any $z \in \mathbb{R}$, if $\Theta^{(s(x(z_t)))}x(z) \leq \psi^{(s(x(z_t)))}$, $\mathcal{A}(z_t) = \mathcal{A}(z)$ and $s(z_t) = s(z)$. By decomposing $x(z) = a + bz$, we have

$$\{\Theta^{(s(x(z_t)))}x(z) \leq \psi^{(s(x(z_t)))}\} = \{\Theta^{(s(x(z_t)))}(a + bz) \leq \psi^{(s(x(z_t)))}\}$$
$$= \{\Theta^{(s(x(z_t)))}bz \leq \psi^{(s(x(z_t)))} - \Theta^{(s(x(z_t)))}a\}$$
$$= \{(\Theta^{(s(x(z_t)))}b)_k z \leq \psi_k^{(s(x(z_t)))} - (\Theta^{(s(x(z_t)))}a)_k \ \text{ for all } k\}.$$

Then, the breakpoint $z_{t+1} > z_t$ at which one node changes its status, i.e., the sign of one inequality is going to be changed, is computed as

$$z_{t+1} = \min_{k:(\Theta^{(s(x(z_t)))}b)_k > 0} \frac{\psi_k^{(s(x(z_t)))} - (\Theta^{(s(x(z_t)))}a)_k}{(\Theta^{(s(x(z_t)))}b)_k}.$$

Hence, we have the result of Lemma 3.

# C    Distribution of naive $p$-value and selective $p$-value when the null hypothesis is true

We demonstrate the validity of our proposed method by confirming the uniformity of $p$-value when the null hypothesis is true. We generated 12,000 null images $x = (x_1, ..., x_n)$ in which $x_i \in [n] \sim \mathbb{N}(0, 1)$ for each case $n \in \{16, 64, 256\}$ and performed the experiments to check the distribution of naive $p$-values and selective $p$-values. From Figure 10, it is obvious that naive $p$-value does not follow uniform distribution. Therefore, it fails to control the false positive rate. The empirical distributions of selective $p$-value are shown in Figure 11. The results indicate our proposed method successfully controls the false detection probability.

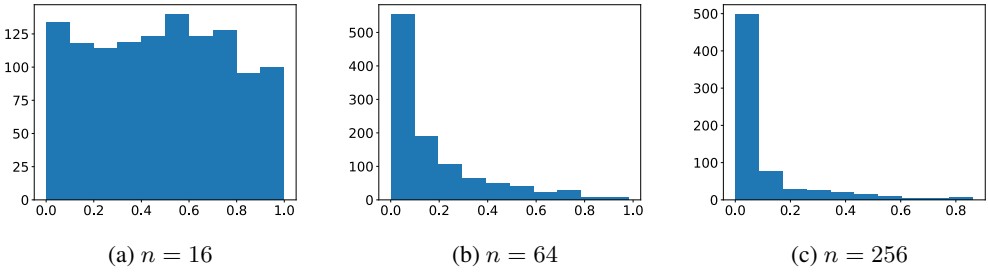

(a) $n = 16$          (b) $n = 64$          (c) $n = 256$

Figure 10: Distribution of naive $p$-value when the null hypothesis is true.

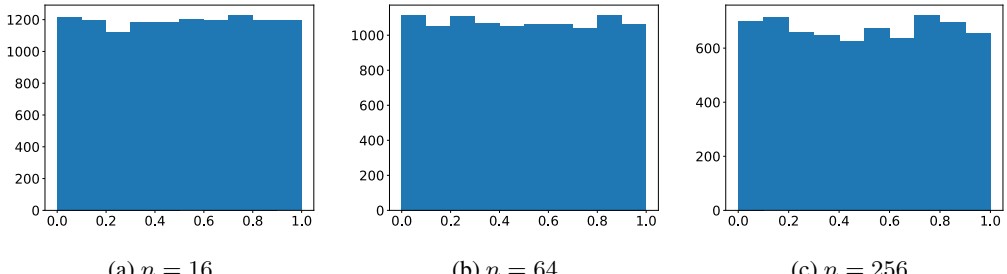

(a) $n = 16$      (b) $n = 64$      (c) $n = 256$

Figure 11: Distribution of selective $p$-value when the null hypothesis is true.

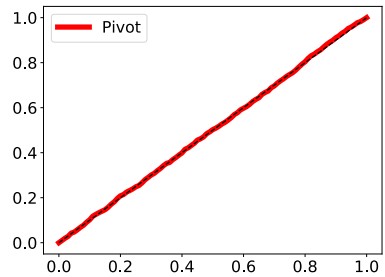
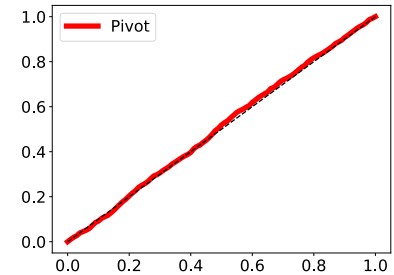

(a) Result for the piecewise linear approximation of sigmoid function.

(b) Result for the piecewise linear approximation tanh function.

Figure 12: Uniform QQ-plot of the pivot.

## D    Examples of piecewise linear linear approximation for non-piecewise linear activation functions.

For simplicity, we consider a simple 3-layer neural network where the activation function at hidden layer is either sigmoid or tanh, the number of input nodes and output nodes are 8, and the number of hidden nodes is 16. Since these functions are non-piecewise linear functions, then we can use the following approximation

$$f(x) = \text{sigmoid}(x) = \begin{cases} 0 & \text{if } x < -4, \\ \frac{1}{8}x + \frac{1}{2} & \text{if } -4 \leq x \leq 4, \\ 1 & \text{if } x > 4. \end{cases}$$

$$f(x) = \tanh(x) = \begin{cases} -1 & \text{if } x < -2, \\ \frac{1}{2}x & \text{if } -2 \leq x \leq 2, \\ 1 & \text{if } x > 2. \end{cases}$$

With the above approximations, we demonstrate that our method still can control the FPR by showing the uniform QQ-plot of the pivot, which is the $p$-value under the null hypothesis, in Figure 12.

In the above example, we used 3 cuts (pieces) to approximate the function. Theoretically, when we increase # cuts, the number of encountered intervals tend to exponentially increase. However, in Figure 13, we show that # encountered intervals still linearly increase in practice.

## E    List of components

Most of components used in DNN for segmentation can be represented as a set of linear inequalities as shown in Table 2.

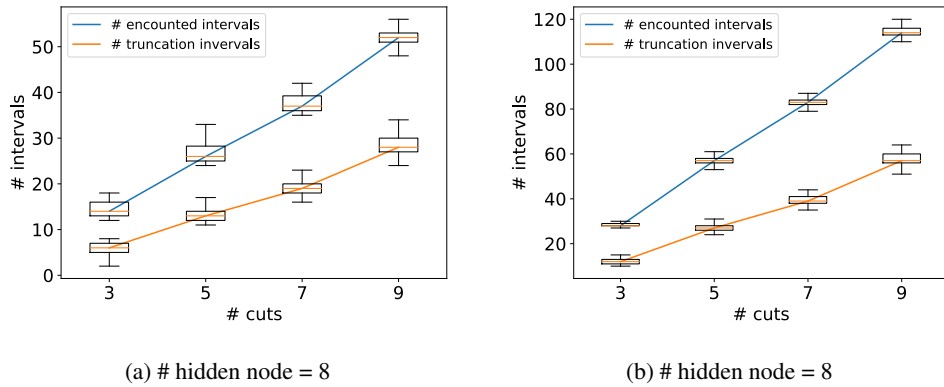

| (a) # hidden node = 8 | (b) # hidden node = 8 |

Figure 13: Demonstration of # encountered and # truncation intervals when increasing # cuts (pieces).

Table 2: Commonly used components in a DNN

| Components | Operation |
|---|---|
| Convolution | linear |
| ReLU activation function | piecewise-linear |
| Max-pooling | comparison |
| Upsampling | linear |
| Thresholding | comparison |
| Concatenate | linear |
| Addition | linear |

## F   Details for experimental setup

We executed the code on Intel(R) Xeon(R) CPU E5-2687W v4 @ 3.00GHz.

The brain image dataset that we used in this paper is extracted from the dataset used in [4], which is available under CC BY-NC-SA 4.0 license.

**Methods for comparison.**   We compare the *proposed method* (homotopy method), which is the main focus of this paper, with the following approaches:

- *Proposed-method-oc (over-conditioning):* This is our first idea of characterizing the conditional data space by additionally conditioning on the observed activeness and inactiveness of all the nodes. The major limitation of this method is its low statistical power due to over-conditioning. The over-conditioning selective $p$-value is computed by (12).
- *Naive method:* This method uses the classical $z$-test to calculate the naive $p$-value.
- *Permutation test:* This too is a well-known technique for computing the $p$-value. The permutation testing procedure is described as follows:
    - Compute the observed test statistic $T^{\mathrm{obs}}$ by applying the trained DNN on $\boldsymbol{x}^{\mathrm{obs}}$
    - For $1 \leftarrow b$ to $B$ ($B$ is the number of permutations which is given by user)
        + $\boldsymbol{x}^{(b)} \leftarrow \mathrm{permute}(\boldsymbol{x}^{\mathrm{obs}})$
        + Compute $T^{(b)}$ by applying the trained DNN on $\boldsymbol{x}^{(b)}$
    - Compute the $p$-value as follows:

$$p_{\mathrm{permutation}} = \frac{1}{B} \sum_{b=1}^{B} \mathbf{1}\{|T^{\mathrm{obs}}| \leq |T^{(b)}|\},$$

where $\mathbf{1}\{\cdot\}$ is the indicator function.

**Definition of power.** Since we conduct statistical testing only when there is a segmentation result discovered by the DNNs, we define the power as follows.

$$\text{Power} = \frac{\# \text{ detected \& rejected}}{\# \text{ detected}}.$$

This definition of power is the same as that in [12] and [8], where $\#$ detected is the number of images on which a segmentation result is obtained by the trained DNN and $\#$ rejected is the number of images whose null hypothesis is rejected by our proposed method.