# OpenReview forum: "Quantifying Statistical Significance of Neural Network-based Image Segmentation by Selective Inference"
_NeurIPS.cc/2022/Conference — NeurIPS 2022 Accept_

### Official Review · Reviewer_W9tv · 2022-07-08

**Rating:** 6
**Confidence:** 4
**Soundness:** 3 good
**Presentation:** 3 good
**Contribution:** 3 good

**Summary:**

This paper concentrates on evaluating the statistical reliability of the segmentation, necessary to develop a plausible statistical inference methodologies. This is difficult in a DNN context. Addressing such a challenge allows to derive exact (non-asymptotic) sampling distribution of the test statistic for obtaining valid statistical inference.
The main goal in this paper t is to conduct the statistical inference in the test phase, specifically, the goal is to quantify the reliability of the segmentation result when a new test image is given to a DNN.


**Questions:**

I consider that this paper is well written, with a clear exposition of the ideas, as well as its analytical development. However, I could no follow all the details provided, as I describe next:
- Perhaps I did not follow the eq.(10) - left side -> x(z) = a+ b*z, I performed some calculations and I obtained  x(z) = a+ b*eta^T * z.
So I wonder if it is possible to obtain this derivation ?

-Seems that in the process of segmentation  we obtain a large number of breakpoints, since the breakpoints are dependent on the number of the neurons in the architecture. Is this a disadvantage of the proposal ?
-By the way, is it possible to have an idea of the  order of magnitude concerning the number of changes in a given segmentation task ?
-The equation in Lemma 3, is computed for each neuron ? If so, I think this is a limitation of the framework
-I understand the authors are concerned to “close” all the operations in a DNN, concerning network operations, activation and pooling functions. However, and following the material provided in the paper, I find a bit short the justifications in Remarks given (page 6). For instance, the universal polling does not fit in Remark 3, that must be, somehow, extended.

(see Junhyuk Hyun, Hongje Seong, Euntai Kim, Universal Pooling -- A New Pooling Method for Convolutional Neural)

- Finally, Although I think that this can constitute an advance in terms of refining the p-value, and consequently, the segmentation confidence, these contributions seem a bit short, when evaluating more complex deep network architectures.  So I may guess that, perhaps this theoretical framework may not be extended to a more drastic situation, where the comçlxity is greatly enlarged.



**Limitations:**

See my questions  above. But fundamentally, extending this framework to more complex architectures is not straightforward.

**Strengths And Weaknesses:**

Strengths
Selective Inference (SI) is known in the community and has been applied in a quite diverse range of applications. However, delving into details in DNN context is a less explored topic. Furthermore, proposed methodologies such as [24] are not applicable in the case of DNN-based segmentation, since the selection event is difficult to be accomplished in this context, as such this  constitutes strength of the proposed approach.
I also think that the background and the theoretical preliminary ingredients of the SI are well written in Section 2.

Weaknesses
Although I think this is a valuable contribution to this research field, I have some questions that perhaps constitute the main weaknesses of the present paper (see my questions below).

---

> ### Author Response · Authors · 2022-07-31
> **Our responses to Reviewer W9tv**
>
> We thank the reviewer for your positive feedback.
>
>
> > I consider that this paper is well written, with a clear exposition of the ideas, as well as its analytical development. However, I could no follow all the details provided, as I describe next: Perhaps I did not follow the eq. (10) - left side $\rightarrow$ $\boldsymbol x(z) = \boldsymbol a+ \boldsymbol b z$, I performed some calculations and I obtained $\boldsymbol x(z) = \boldsymbol a+ \boldsymbol \beta^\top  z$. So, I wonder if it is possible to obtain this derivation?
>
> Please kindly refer to Appendix A (Proof of Lemma 1) in which we provided the derivation of $\boldsymbol x(z) = \boldsymbol a+ \boldsymbol b z$. Regarding the $\boldsymbol x(z) = \boldsymbol a + \boldsymbol \beta^\top  z$ that the reviewer derived, it is a little bit strange to us because we do not use any notation 'beta'. We guess that the reviewer means $\boldsymbol x(z) = \boldsymbol a + \boldsymbol \eta z$ ('beta' $\rightarrow$ 'eta' and no transpose). If so, our answer is Yes. We can obtain $\boldsymbol x(z) = \boldsymbol a + \boldsymbol \eta z$ when $\Sigma$ is identity matrix and $||\boldsymbol \eta ||_2 = 1$.
>
>
> > Seems that in the process of segmentation we obtain a large number of breakpoints, since the breakpoints are dependent on the number of the neurons in the architecture. Is this a disadvantage of the proposal?
>
> No, it is not. Actually, in the proposed method, we introduced the homotopy approach  to circumvent the enumeration of all possible combinations of activeness of the hidden nodes, which is exponentially increasing. If we naively consider all possible combinations of activeness of the hidden nodes, it would only be applicable to an overly simple toy network with about 10 hidden nodes. On the other hand, by doing parametrized line search with the proposed method, we only need to consider the smaller number of breakpoints that actually affects the computation of $\mathcal Z$.
>
>
> > By the way, is it possible to have an idea of the order of magnitude concerning the number of changes in a given segmentation task?
>
> As we discussed in the complexity paragraph at the end of Sec. 3, in the worst case, the complexity still grows exponentially and it is a common issue in other homotopy applications such as Lasso regularization path. However, fortunately, it has been well-recognized that this worst case rarely happens in practice and the actual cost is known to be nearly linear order. Similarly, in the proposed homotopy method for SI, it is also evident from our experimental results that the number of breakpoints is almost linearly increasing (Fig. 6).
>
> > The equation in Lemma 3 is computed for each neuron? If so, I think this is a limitation of the framework.
>
> No, it is not. Lemma 3 is computed for each breakpoint where each breakpoint corresponds to a particular combination of the activenesses of the neurons in a network. Therefore, the number of times we need to compute the equation in Lemma 3 equals the number of breakpoints.
>
> > I understand the authors are concerned to “close” all the operations in a DNN, concerning network operations, activation and pooling functions. However, and following the material provided in the paper, I find a bit short the justifications in Remarks given (page 6). For instance, the universal polling does not fit in Remark 3, that must be, somehow, extended.
> (see Junhyuk Hyun, Hongje Seong, Euntai Kim, Universal Pooling -- A New Pooling Method for Convolutional Neural)
>
> We agree with the reviewer's comment. In the revised version, we will provide a list of popular components that we can handle as well as the list of the ones that our method currently does not support to supplement the remarks and make the discussion clearer.
>
>
> > Finally, although I think that this can constitute an advance in terms of refining the $p$-value, and consequently, the segmentation confidence, these contributions seem a bit short, when evaluating more complex deep network architectures. So I may guess that, perhaps this theoretical framework may not be extended to a more drastic situation, where the complexity is greatly enlarged.
>
> The proposed method can be applied to a class of networks that is represented as a piecewise-linear function (Sec. 3), and for this class piecewise linear networks, theoretical validity is guaranteed regardless the size of the dataset and the complexity of the network. It is known that a large class of deep networks, including some of the SOTA image segmentation networks can be represented (or approximated with sufficient accuracy) as a piecewise linear function (see, e.g., Bunnel et al. A unified view of piecewise linear neural network verification. NeurIPS 2018). Namely, there is no technical limitation in applying our proposed method to SOTA structures. However, it requires a considerable amount of implementation effort. In the revised manuscript, we will carefully discuss the possibilities for application to more complex networks.

---

### Official Review · Reviewer_BVA9 · 2022-07-11

**Rating:** 6
**Confidence:** 3
**Soundness:** 3 good
**Presentation:** 4 excellent
**Contribution:** 3 good

**Summary:**

The paper proposes a method that extends previous work on conditional selective inference (SI) for statistical significance quantification in segmentation models to deep neural networks with affine operations and piecewise linear activation functions (the previous method was applicable to threshold- and graph-cut-based algorithms only). The core idea of this approach is to correct for the segmentation bias and perform SI on a conditional subspace of images that yield the same segmentation result as that actually observed. To identify this subspace, the paper proposes to perform search on a line parametrized by a scalar $z$ and reduces the core problem to finding the set of $z$'s compatible with the actual image ("truncation region"). To do this, it proposes to overcondition on "selected pieces" of the activations of the network nodes, shows that this can be expressed in the form of linear inequalities, and that multiple overconditioning cases can be combined with a homotopy method by finding "breakpoints" where exactly one network node changes its value. The method is experimentally evaluated on synthetic random data and on real images of brain tumors, using a simple conv-pooling network.


**Questions:**

- The paper generally presents the method as "efficient" in a few places, and on page 8 mentions that the homotopy method can be exponential in the worst case, but tends to be linear in practice. I think it would be helpful for practitioners to also mention explicitly in the text that the linear complexity has a high "constant factor" in the form of neural network inference (obvious in Alg. 2, but a bit hard to see from just skimming the paper).
- nit: "Therefore, all of the operations that only operate in the training process such as Dropout or Batch Normalization"; this is not strictly true about dropout and BN, as these operations also affect inference (albeit in a way that is compatible with the claims in the paper and does break the assumption of the operations being affine)
- in ref [24], it is assumed that $\Sigma$ can be estimated from a null image known not to contain any "foreground" pixels; It would be useful to mention how it was estimated to generate the results in Fig. 7.
- nit: in Alg. 2, the initialization of T should be {z_t} instead of $z_t$


**Limitations:**

No concerns.

**Strengths And Weaknesses:**

The paper is very well written, cites the relevant literature, and includes a brief introduction to SI for simple segmentation methods, summarizing the results from ref [24]. The reduction of the data subspace to 1D is based on prior results from ref [16], which the present paper acknowledges. The major contribution of the present work is showing how overconditioning and the homotopy method allow the application of SI to neural network-based segmentation models, and the demonstration of its effectiveness on sample synthetic and real data. This is a novel result and should be of significant interest.

The paper is generally upfront about the limitations (affine operations only, hard to approximate softmax), but could be a bit more clear about the computational complexity, which is important for real-life applications.

It is also commendable that the authors included the code necessary to reproduce all the experiments and the figures.

---

> ### Author Response · Authors · 2022-07-31
> **Our responses to Reviewer BVA9**
>
> We thank the reviewer for your positive comments.
>
> > The paper generally presents the method as "efficient" in a few places, and on page 8 mentions that the homotopy method can be exponential in the worst case, but tends to be linear in practice. I think it would be helpful for practitioners to also mention explicitly in the text that the linear complexity has a high "constant factor" in the form of neural network inference (obvious in Alg. 2, but a bit hard to see from just skimming the paper).
>
> Based on your suggestion, we will update the complexity paragraph in the revised paper as follows:
>
> "In the literature of homotopy method (a.k.a. parametric programming), it is known that the actual computational cost differs significantly from the worst case. A well-known application of the homotopy method in the ML community is the Lasso regularization path, which also has the worst-case computational cost on the exponential order of the number of features, but the actual cost is known to be nearly linear order. Similarly, in the proposed homotopy method for SI, it is also evident from our experimental results that the number of breakpoints is almost linearly increasing in practice (Fig. 6)."
>
>
> > nit: "Therefore, all of the operations that only operate in the training process such as Dropout or Batch Normalization"; this is not strictly true about dropout and BN, as these operations also affect inference (albeit in a way that is compatible with the claims in the paper and does break the assumption of the operations being affine)
>
> As the reviewer pointed out, our description of BN was misleading. In Sec. 5 of the revised version, we discuss the possible difficulty of the piecewise-linear approximation of BN as a limitation of the proposed method.
>
> > In ref [24], it is assumed that $\Sigma$ can be estimated from a null image known not to contain any "foreground" pixels; It would be useful to mention how it was estimated to generate the results in Fig. 7.
>
> In the revised paper, we will add the following discussion in numerical results paragraph of Sec. 4.
>
> "Regarding the last plot (estimated sigma) in Figure 7, since we want to empirical check the performance on FPR control when breaking the assumption of $\Sigma$ (i.e., $\Sigma$ is known or estimated from independent data), we set $\Sigma = s^2  I $ where $s^2$ is the empirical (sample) variance and $I$ is the identity matrix."
>
> > nit: in Alg. 2, the initialization of T should be {z_t} instead of $z_t$
>
> We will correct it.

---

> > ### Comment · Reviewer_BVA9 · 2022-08-08
> > **Thanks**
> >
> > Thank you for the response -- I think the changes you propose will be very helpful. I'd also recommend being a bit more explicit about the required number of forward passes of the network, in addition to the discussion of expected time complexity of your method.

---

> > > ### Author Response · Authors · 2022-08-09
> > > **Our response to additional suggestion of Reviewer BVA9**
> > >
> > > Thank you for your additional suggestion. In the revised paper, we will additionally provide the following discussion:
> > >
> > > "At step 4 of Algorithm 2, we apply a trained DNN to the parametrized data $\boldsymbol x(z_t)$ at any breakpoints $z_t$. Therefore, the number of forward passes of the network is equal to the number of breakpoints."

---

### Official Review · Reviewer_6Pq7 · 2022-07-19

**Rating:** 5
**Confidence:** 3
**Soundness:** 3 good
**Presentation:** 3 good
**Contribution:** 3 good

**Summary:**

This paper studies the statistical reliability of segmentation results obtained by Neural Network-based Image Segmentation. The authors developed a conditional selective inference (SI) framework to compute exact valid p-values for the segmentation results. The experimental results on the brain image dataset demonstrate the effectiveness of the proposed method.

**Questions:**

Please refer to the Question in the Weaknesses.

**Limitations:**

Yes

**Strengths And Weaknesses:**

[Strengths]
+ The studied problem is interesting and important.
+ Compared to the Permutation Test, the proposed methods could bring significant improvements.

[Weaknesses]
- The authors just conduct the experiments on a small dataset, which makes it difficult to evaluate the robustness of the proposed method.
- The evaluated NNs model is somehow simple.  There are many SOTA image segmentation / Semantic segmentation methods, e.g., Deeplabv3+, and datasets, e.g., Pascal VOC. Could you evaluate these segmentation methods on the dataset with the proposed method?
- What’s the relationship between Uncertainty Estimation for Semantic Segmentation and statistical reliability?

---

> ### Author Response · Authors · 2022-07-31
> **Our responses to Reviewer 6Pq7**
>
> We thank the reviewer for your feedback.
>
> > The authors just conduct the experiments on a small dataset, which makes it difficult to evaluate the robustness of the proposed method. The evaluated NNs model is somehow simple. There are many SOTA image segmentation / Semantic segmentation methods, e.g., Deeplabv3+, and datasets, e.g., Pascal VOC. Could you evaluate these segmentation methods on the dataset with the proposed method?
>
> The proposed method can be applied to a class of networks that is represented as a piecewise-linear function (Sec. 3), and for this class piecewise linear networks, theoretical validity is guaranteed regardless the size of the dataset and the complexity of the network. It is known that a large class of deep networks, including some of the SOTA image segmentation networks (e.g., U-Net, SegNet) can be represented (or approximated with sufficient accuracy) as a piecewise linear function (see, e.g., Bunnel et al. A unified view of piecewise linear neural network verification. NeurIPS 2018). Namely, there is no technical limitation in applying our proposed method to SOTA structures. However, it requires a considerable amount of implementation effort. Our main contribution is to first introduce a theoretically valid statistical hypothesis test framework (in the sense that the probability of false positives is guaranteed to be less than the significance level $\alpha$) in the context of neural network-based segmentation. As the reviewer pointed out, experiments on SOTA networks are important, but even without demonstration on SOTA networks, we believe that advancing the theoretical understanding of deep learning would be valuable to the ML community.
>
>
> > What’s the relationship between Uncertainty Estimation for Semantic Segmentation and statistical reliability?
>
> Statistical reliability, which we mainly focus on in this study, is one of several approaches for uncertainty quantification in ML (e.g., statistical reliability, causality, or robustness). The main target domain of this study is the segmentation problem for detecting abnormal regions, as often studied in medical image diagnosis. In the segmentation problem for detecting abnormal regions, it is important to quantitatively evaluate the reliability of the detected abnormal regions, and especially in the life science field, it is necessary to guarantee the statistical significance of the abnormality using $p$-values. On the other hand, the interpretation of the $p$-value is not always clear in semantic segmentation for natural images.

---

### Meta-Review · Area_Chair_HNQi · 2022-09-10

**Recommendation:** Accept
**Confidence:** Certain

**Metareview:**

This paper proposes a conditional selective inference based technique for quantifying the statistical significance of an image segmentation generated by a DNN. Estimating the statistical significance of a DNN generated image segmentation is an important problem in medical image analysis, and the technique proposed extends the applicability of selective inference beyond the previously applicable domains of thresholding or graph partitioning based segmentation methods to segmentations generated by DNNs with piecewise linear activation functions.

While all reviewers found the proposed method to be of value to the community at large, they commented on the limited evaluation of the method. The method has only been applied to simple segmentation networks, and not state of the art architectures, further it has only been applied to a limited set set of datasets, suggesting the practical challenges with scaling this technique to realistic networks and datasets. Regardless, the reviewers unanimously recommend acceptance, and I hope this work will stimulate further practical progress in this important area.



**Award:**

No

---

### Decision · Program_Chairs · 2022-09-14

Accept